# Recent Developments in Luffa Natural Fiber Composites: Review

**Mohamad Alhijazi [1], Babak Safaei [1,*], Qasim Zeeshan [1], Mohammed Asmael [1], Arameh Eyvazian [2,3] and Zhaoye Qin [4,*]**

[1] Department of Mechanical Engineering, Eastern Mediterranean University, Famagusta 99628, North Cyprus via Mersin 10, Turkey; mohamadalhijazi@gmail.com (M.A.); qasim.zeeshan@emu.edu.tr (Q.Z.); mohammed.asmael@emu.edu.tr (M.A.)

[2] Institute of Research and Development, Duy Tan University, Da Nang 550000, Vietnam; arameheyvazian@duytan.edu.vn

[3] Faculty of Electrical—Electronic Engineering, Duy Tan University, Da Nang 550000, Vietnam

[4] Department of Mechanical Engineering, Tsinghua University, Beijing 100084, China

* Correspondence: babak.safaei@emu.edu.tr (B.S.); qinzy@mail.tsinghua.edu.cn (Z.Q.)

**Abstract:** Natural fiber composites (NFCs) are an evolving area in polymer sciences. Fibers extracted from natural sources hold a wide set of advantages such as negligible cost, significant mechanical characteristics, low density, high strength-to-weight ratio, environmental friendliness, recyclability, etc. *Luffa cylindrica*, also termed luffa gourd or luffa sponge, is a natural fiber that has a solid potential to replace synthetic fibers in composite materials in diverse applications like vibration isolation, sound absorption, packaging, etc. Recently, many researches have involved luffa fibers as a reinforcement in the development of NFC, aiming to investigate their performance in selected matrices as well as the behavior of the end NFC. This paper presents a review on recent developments in luffa natural fiber composites. Physical, morphological, mechanical, thermal, electrical, and acoustic properties of luffa NFCs are investigated, categorized, and compared, taking into consideration selected matrices as well as the size, volume fraction, and treatments of fibers. Although luffa natural fiber composites have revealed promising properties, the addition of these natural fibers increases water absorption. Moreover, chemical treatments with different agents such as sodium hydroxide (NaOH) and benzoyl can remarkably enhance the surface area of luffa fibers, remove undesirable impurities, and reduce water uptake, thereby improving their overall characteristics. Hybridization of luffa NFC with other natural or synthetic fibers, e.g., glass, carbon, ceramic, flax, jute, etc., can enhance the properties of the end composite material. However, luffa fibers have exhibited a profuse compatibility with epoxy matrix.

**Keywords:** luffa natural fiber composites; mechanical properties; physical properties; thermal properties; chemical properties; morphological properties

## 1. Introduction

Fiber-reinforced composites are becoming significantly popular in various engineering fields due to their low density as well as their remarkable mechanical characteristics. Composite materials' properties are based on the selected components, viz., matrix and fibers [1–3]. Hitherto, the utilized matrices and fibers are generally obtained from petroleum origins. Although they possess attractive properties, the aforementioned composite materials are restricted from being used for long periods, yet can endure regular environmental conditions for tens of years [4–9]. Moreover, composite recycling and reprocessing methods are unavailable. The manufacturing of most synthetic fibers has several

environmental effects, as it requires high power, is toxic for humans, as well as may deplete the ozone layer, cause global warming and eutrophication [10–13]. The increased attentiveness to environmental matters has augmented the search for an alternative natural source in order to increase the utilization of renewable materials, reduce waste production, boost recycling, and so on [14,15].

Hence, numerous scientists and engineers tend toward green materials that can enhance the products' environmental aspects [13,16–19]. Thus, materials extracted from natural resources gained more attention as an alternative to synthetic fibers in composite materials. Natural fibers (NFs) are found in many parts of a plant, e.g., fruit, bast, leaf, trunk, roots, and so on. Throughout the decades, these natural fibers have been widely utilized in countless applications due to their advantages, as they are strong, lightweight, tough, recyclable, biodegradable, abundant in nature, and have negligible cost and low density [20–27]. Additionally, their environmental advantages include decreased respiratory and dermal irritation, improved energy consumption, less wear and abrasion on tools, and minimal health hazards. Natural fiber composites (NFCs) have been significantly involved in various engineering fields such as automotive, marine, sports gear, construction, and aerospace [17,28–34].

NFs, also named lignocellulosic fibers, comprise wheat straw, sea grass, softwood kraft, sisal, rice husk, ramie, rachis, pineapple, oil palm, date palm, jute, hemp, flax, coconut, coir, cotton, banana, bamboo, abaca, mesta, roselle, oat, maize, sabia, kapok, etc. [35–43]. Figure 1 shows the main fiber categories. Besides the prosperous characteristics of natural fibers, they have a few disadvantages due to their hydrophilic behavior. However, there are several methods that can reduce these drawbacks, like adding coupling agents and fiber treatment. Luffa fruit comprise lightweight natural fibers that have the potential to be utilized in reinforcing lightweight composites due to their polyporous structure, abundance, cheap price, as well as their surface morphology, which can provide a good adhesion with the matrix. The main focus of this review paper is to compile, compare, and summarize the research on luffa natural fiber composites (LNFCs) by considering the physical, morphological, mechanical, chemical, electrical, and thermal properties, in addition to acoustic, water absorption, x-ray diffraction, differential scanning calorimetric, thickness swelling, and so on.

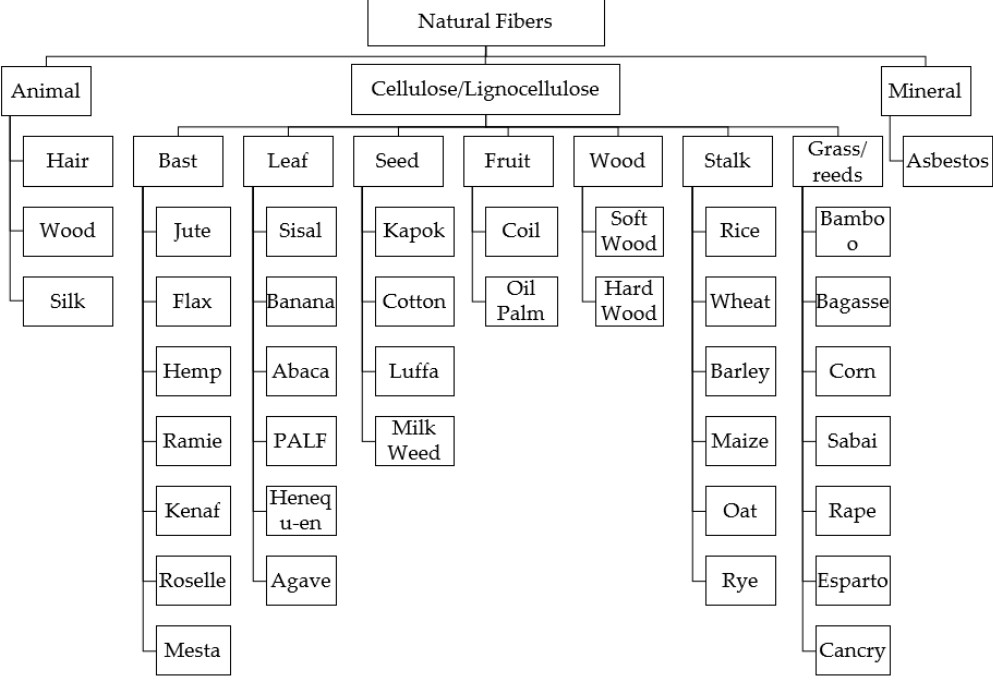

**Figure 1.** General classifications of natural fibers.

## 2. Luffa and Its Composite Materials

Luffa is a category of the Cucurbitaceae family (cucumber), its ripe fruits are utilized as natural cleaning sponges, while its immature fruits are consumed as vegetables. It is spread from south Asia to east and central Asia. Luffa vegetables are widespread in Vietnam and China. Figure 2 shows the mature luffa fruit and its fiber structure. Luffa fibers comprise significant toughness, strength, and stiffness, similar to the ones observed in various metals with same density ranges [44].

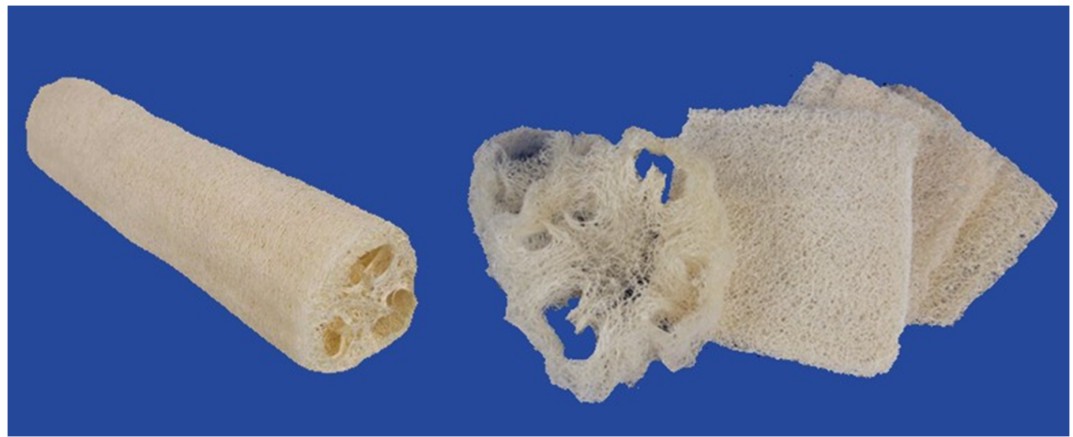

**Figure 2.** Luffa and its internal structure.

Luffa chemical composition mostly consists of lignin and hemicellulose/cellulose, as well as includes some inorganic elements like glycosides, polypeptides, amino acids, proteins, and so on [45–47]. However, the hemicellulose content ranges between 8% and 22%, lignin content is between 10% and 23%, and cellulose content is between 55% and 90%. Table 1 shows the physical and chemical properties of luffa. At the early stage of luffa growth, its cellular structure begins with numerous single fibers and turn into fibrous mat at the end [48].

**Table 1.** Physical properties and chemical composition of luffa fibers [49–51].

| Physical Properties | | | | Chemical Composition | | | |
|---|---|---|---|---|---|---|---|
| Density (gm/cm$^3$) | Diameter (μm) | Aspect ratio | Micro fibrillar angle (°) | Cellulose (%) | Lignin (%) | Hemi cellulose (%) | Ash (%) |
| 0.56−0.92 | 270 ± 20 | 340 ± 5 | 12 ± 2 | 63.0 ± 2.5 | 11.69 ± 1.2 | 20.88 ± 1.4 | 0.4 ± 0.10 |

As Table 1 shows, luffa density varies from 0.56 to 0.92 g/cm$^3$, it has an average diameter of ~270 μm, and its microfibrillar angle is around 12°. The chemical composition of luffa consists of 63% cellulose, 20.88% hemicellulose, 11.69% lignin, and 0.4% ash. It is worth mentioning that in addition to its use as a vegetable and cleaning sponge, luffa is also utilized in Chinese medication, military filters, and shock absorbers [52].

### 2.1. Fiber Treatment

Water absorption and moisture retention harm the fiber/matrix adhesion in composite materials. Moreover, NFs have high moisture absorption properties as they are naturally hydrophilic. Such properties cause a reduction in bond strength, and thus matrix and fibers detach from each other. Hence, these composite materials exhibit negligible mechanical characteristics in wet environments [53]. Therefore, treating an NF with a convenient chemical solution can influence its chemical composition, remove surface impurities, as well as reduce its water absorption character. Table 2 shows the treatments applied to luffa fibers in LNFC studies.

**Table 2.** Luffa fiber treatments.

| Treatment | Reference |
|---|---|
| Sodium Hydroxide (NaOH) | [25,44,49–51,54–76] |
| Hydrogen Peroxide ($H_2O_2$) | [27,54,55,65–67] |
| Acetic Acid ($CH_3COOH$) | [27,54,55] |
| Carbamide $CO(NH_2)$ | [55] |
| Methacrylamide | [60,69] |
| Benzoyl Chloride Permanganate ($KMnO_4$) | [51,75] |
| Acetic Anhydride, and Acetone | [65] |
| Furfuryl Alcohol followed by oxidation (sodium chlorite + acetic acid) | [50] |
| $CaCl_2$, $H_2SO_4$, and $Na_2HPO_4$ | [57] |
| Hypochlorite (NaClO) | [58] |
| Ethanol, BTDA Dianhydrides | [64] |
| HCl | [56] |
| Chlorine Bleach | [46,77] |
| CalciumPhosphate and Calcium Carbonate | [78] |
| $CaOH_2$ and Silane | [71] |
| Thermo-mechanical treatment and thermo-hydromechanical treatment | [79] |
| Heat treatment | [80] |

Sodium hydroxide (NaOH)/alkaline treatment evidenced its capability in improving luffa fibers' microstructure by changing its chemical composition as well as removing all impurities [49,59–63,70]. Treating luffa fibers with 4% NaOH at 120 °C for 3 h revealed the highest fiber crystallinity index and, in addition, combined chemical treatments switched luffa from a mat into a filament structure [67]. Contrary to other chemical solutions, methacrylamide treatment caused a serious deterioration in luffa fiber integrity [69]. Mixing NaOH with other solutions like $CH_3COOH$ can drastically improve LNFC mechanical performance as well as significantly decrease its water absorption; in contrast, mixing with $H_2O_2$ deteriorated its mechanical characteristics [55]. The tensile strength of LNFC created with HCl treated fibers was lower than that of LNFC treated with alkaline [56]. Cyanoethylating and acetylation improved fiber/matrix adhesion, resulting in an enhancement in mechanical characteristics [65,66]. Furfuryl alcohol followed by oxidation treatment revealed higher performance compared to alkaline, where it improved the surface structure and reduced hemicellulose, lignin, and wax quantities [50]. As shown in Table 2, sodium hydroxide was mostly utilized to chemically treat luffa fibers, followed by hydrogen peroxide and acetic acid.

*2.2. Matrices Selected for LNFCs*

Composites have a combination of fibers and matrix properties, and in addition to matrix properties, they behave as a structure that holds all fibers together, as well as a protection from the surrounding environment (water, heat, etc.) [1,81–83]. Thus, studying the performance of a new NFC involves choosing a suitable matrix that exhibits good properties with a considerable interaction with the selected NF. Several studies investigated luffa as a pure mat (without a matrix); however, others studied different thermoplastics and thermosets like epoxy, polyester, resorcinol-formaldehyde, vinyl ester, and so on [84–86]. Matrices considered in recent LNFC studies are listed in Table 3.

**Table 3.** Thermoplastics and thermosets used in luffa natural fiber composite (LNFC) development.

| Matrix | Reference |
| --- | --- |
| Epoxy | [25,44,49–51,54,59,61–63,72–75,84,87–93] |
| Polyester | [24,56,60,65,66,70,71,84,94–96] |
| Resorcinol-formaldehyde | [57,78,97] |
| Polylactic acid | [58,80] |
| Bio-based polyethylene (HDPE) | [98] |
| Vinyl ester | [7,64] |
| Polyurethane foam | [45] |
| Polyurethane (PU) | [99] |
| Polypropylene | [68] |
| Geopolymer | [100] |
| Pre-gelatinized cassava starch | [101] |
| Eva resin | [102] |
| Bismuth nitrate pentahydrate ($Bi(NO_3)_3 \cdot 5H_2O$) and potassium iodide | [76] |
| Concrete | [103] |

LNFC studies have involved diverse polymeric matrices with different weight ratios (fiber volume fraction), which ranged from 2 wt.% to 50 wt.%, however, the most common weight composition was 30 wt.% [24,49,50,62,70]. Although the majority of studies considered luffa as rectangular mat, some utilized it as randomly chopped fibers between 2 mm and 6 cm. As clearly shown in Table 3, epoxy resin was selected most often in the LNFC area, followed by polyester and resorcinol-formaldehyde, which is due to matrix properties as well as matrix/fiber compatibility.

## 3. Characteristics of LNFCs

The characteristics of a natural fiber composite are essential for ensuring its appropriateness for being implemented in industrial sectors [104–107]. These include mechanical properties such as tensile strength, flexural strength, impact, compression, hardness, etc., and physical properties like water absorption, density, and so on, taking into account its behavior towards chemical solutions and thermal changes [40,43,108]. This section includes a summary and comparison of findings revealed in published LNFC researches.

### 3.1. Morphology of LNFCs

One of the most important factors in NFC development is studying its microstructure in order to determine its fracture type, fiber surface morphology (especially to figure out the effect of a treatment), crystallinity index, and so on. Moreover, studying the morphology of an LNFC identifies the degree of compatibility of these fibers with the selected matrix, and thereby specifies the condition of fiber-to-matrix adhesion and manifests the internal flaws. Usually, morphological analyses are conducted on the fracture surface of a failed LNFC specimen, which helps to define the failure nature (brittle or ductile). Scanning electron microscopy (SEM), transmission electron microscopy, polarized light microscopy, X-ray diffraction [109] and other tests were utilized in several studies to analyze the morphology of luffa mat/fiber and LNFC [109–114].

### 3.1.1. Microscopy

Diverse microscopic techniques have been used for luffa fibers and LNFCs, such as polarized light microscopy (PLM), scanning electron microscopy (SEM), field emission scanning electron microscopy (FESEM), optical microscopy, etc. However, researches have considered scanning electron microscopy most often for inspecting the fiber surface morphology, voids presence, fiber/matrix interaction, and so on. Few researches scanned pure luffa samples [55,67,69], while others involved the following matrices: epoxy [44,50,51,59,61–63,89,91], polyester [60,66,96], polypropylene [68], vinyl ester [64],

geopolymer [100], pre-gelatinized cassava starch [101]. SEM micrographs of untreated and treated fibers are shown in Figures 3 and 4; FESEM micrographs of treated luffa/epoxy are illustrated in Figure 5.

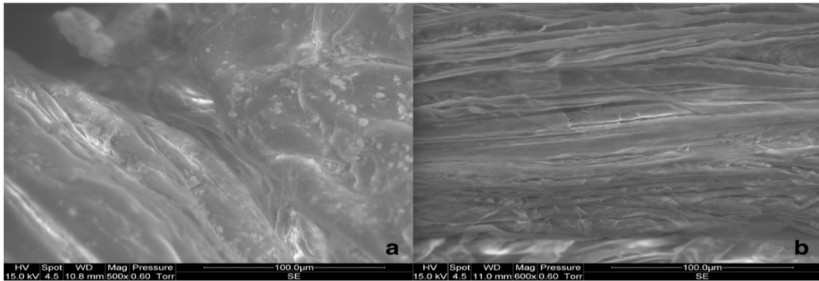

**Figure 3.** SEM micrographs of (**a**) untreated luffa fibers; (**b**) alkali-treated luffa fibers. Adapted with permission from ref. [67]. Copyright 2020, Elsevier.

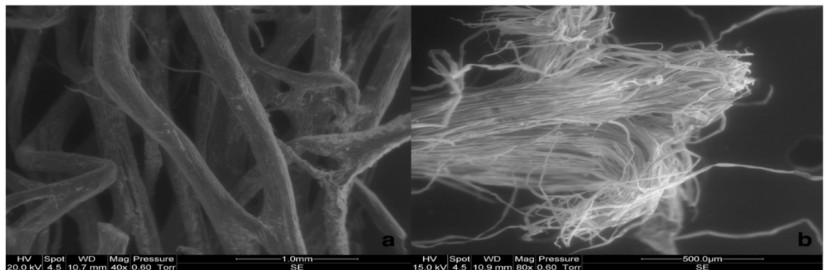

**Figure 4.** SEM micrographs of (**a**) morphology of untreated luffa fibers; (**b**) fibrillary structure of treated fibers. Adapted with permission from ref. [67]. Copyright 2020, Elsevier.

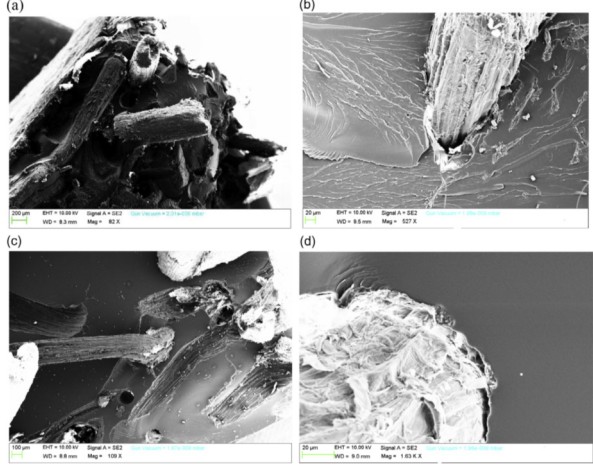

**Figure 5.** Field emission SEM of treated luffa/epoxy NFC (**a**) Fracture surface of short fibers LNFC (82×), (**b**) Luffa fiber cracked side (527×), (**c**) fracture surface of mat LNFC (100×) and (**d**) Luffa fiber cracked side (100k×). Adapted with permission from ref. [59]. Copyright 2020, Elsevier.

SEM contributes to showing the fibers' surface roughness as well as their fibrous structure. Normally, the outer surface of lignocellulosic fibers consists of a waxy and fatty layer with salt-like solids. Luffa fibers have a flake-like structure and are highly capable of enduring tensile forces. Selecting these fibers as a reinforcement can enhance the composite's strength [44]. Extraction treatment reduced the amount of solids covering the fibers' surface by decreasing the roughness rate [64]. From SEM micrographs, Anbukarasi et al. [59] observed that particle fibers and short fibers had a promising behavior in LNFC. In addition, considering luffa as a mat contributes to restricting the fibers from

pulling out from the matrix due to its network structure, which thereby improved the mechanical properties such as tensile strength, flexural, impact, etc. [62]. Chemically treating luffa fibers reduced the hydroxyl groups' number as well as increased the fibers' surface roughness [83]. Hence, fiber treatment improved their bonding with the matrix and reduced the voids content [68,115,116].

### 3.1.2. X-ray Diffraction

Generally, an X-ray diffraction test (XRD) helps in evaluating the crystallographic nature of natural fiber composites like LNFC, as well as determining the impact of chemical treatments on the crystalline nature. Recent research implemented XRD to study LNFCs with an epoxy matrix [51], polyester matrix [65,66,96], resorcinol-formaldehyde [57,78], geopolymer [100], polylactic acid [58], and pure luffa fibers [27,55,67,69].

LNFC cell walls contain mainly lignin, hemicellulose, and cellulose. Hemicellulose and lignin are amorphous, while cellulose includes crystalline as well as amorphous phases [51]. Almost all researches obtained maximum crystallinity values in a range between 20° and 25° of 2-Theta, while the amorphous peaks were exhibited between 14° and 16° [51]. By increasing the fiber volume fraction in LNFC, the crystallite size and crystallinity decreased, which evidenced a remarkable bone bonding as well as bone implant [78]. Chemically treating luffa fibers increased the cellulose crystallinity index (a value of around 60%), which can result in a simultaneous increase in that of the end LNFC [57]. Various chemical treatments were able to improve the crystallinity index of luffa fibers and showed a range of values from 59% to 63% [69].

### *3.2. Physical Properties*

Developing a new natural fiber composite like LNFC is critical in terms of its physical characteristics [117,118]; for example, its density should be compared with materials from the same category, its thickness swelling has to be inspected in order to define its dimensional stability, and most importantly, as a natural fiber composite, its water absorption must be examined since it may lead to a drastic property degradation. Since NFCs are being utilized in the industry for sound insulation purposes, the acoustic behavior of luffa fibers is high in importance, as well [119,120].

### 3.2.1. Density

Basically, the density of LNFC is measured by weighing the cured samples and measuring their dimensions, then dividing the obtained mass by volume. Densities of LNFCs with various matrices are listed in Table 4.

**Table 4.** Density of luffa natural fiber composites.

| Resin/Coupling Agent/Hardener | Fiber Treatment | Fiber Size/Shape | Fiber Composition Vf/wt.% | Hybrid/Filler | Density g/cm$^3$ | Reference |
|---|---|---|---|---|---|---|
| Epoxy/Hardener HY951 Epoxy/hardener K-6 | NaOH | Rectangular mat 2–5 mm | 19.87–30.86% Vf 3.2–9.6 wt.% | Glass fiber | 1.009–1.297 1.1142–1.1501 | [91] [25] |
| Formaldehyde | | Mat in medium density fireboard | | Pine, beech, and oak | 0.717–0.721 | [97] |
| Geopolymer (metakaolin activated with sodium silicate and sodium hydroxide solutions) | | Fiber's D: 200 μm. L: 160 mm | 10 vol.% | | 1.38 | [100] |

The density of LNFCs range between 0.7 and 1.38, depending the selected matrix (as shown in Table 4). Besides the notable improvement in the mechanical properties of SK-geopolymer after the addition of luffa fibers as a reinforcement, the end composite material revealed a 1.38 g/cm$^3$ density lower than that of pure geopolymer (1.5 g/cm$^3$) [100]. Regarding LNFC with an epoxy matrix, increasing the fiber volume fraction by up to 9.6% decreased the overall density by 3.12% [25]. Average

density of luffa/formaldehyde NFC ranges between 0.719 to 0.721 g/m$^3$; however, the density of this NFC decreased by increasing fiber content [97]. Moreover, the observed density reduction emphasized the light weight of LNFC.

### 3.2.2. Water Absorption

Water absorption is one of the main concerns in utilizing NFCs in several areas. This property is usually tested by immersing NFC specimens in distilled water for a period of time at room temperature, then removing the specimens and wiping the remaining water droplets from their surfaces [121,122]. Hence, the weight of these samples is measured and compared with their initial weight in order to calculate the weight change ratio [123,124]. Water absorption property was tested for LNFCs with several matrices such as: epoxy [49,50,59,62,63], polyester [24,60,66,95], polypropylene [68], vinyl ester [64], polylactic acid [80], pre-gelatinized cassava starch [101], as well as pure luffa bundles [27,55]. Since luffa fibers are hydrophilic in nature, their addition into a matrix increased the overall water absorption of the NFC due to the existence of homocellulose and pectin. Therefore, water absorption of LNFC increased by increasing the fiber volume fraction [49]. Furthermore, LNFC with treated as well as untreated fibers showed a quick weight increase in the first week of its immersion in water. Yet the water turned blurry, which indicated that the polymeric matrix was losing some of its components [60]. However, untreated LNFC exhibited around 14% higher water uptake, compared to treated LNFC, which indicated that the latter had a better compatibility with the matrix [59,63]. Additionally, void content and density of the material significantly affected its water absorption [24].

### 3.2.3. Thickness Swelling

Thickness swelling is a critical property in natural fiber composites, it describes the dimensional stability of these composites after absorbing moisture. However, the swelling rate is low throughout the early moisture absorption stages of a natural fiber composite made with polymeric matrix, due to the viscoelastic behavior of this polymer. Hence, reaching a high swelling rate leads to the deterioration of the mechanical properties such as tensile strength, compression, impact, bending, etc. To date, research on thickness swelling of LNFC is still limited. Akgül et al. [97] observed that the thickness swelling of luffa/formaldehyde NFC increased by increasing luffa layers in the sample, whereas samples with three luffa layers exhibited peak thickness swelling percentage.

### 3.2.4. Acoustic Resistance

Enhancing the environmentally friendly composites, LNFCs have a solid potential to be involved in several vibration and sound isolation utilizations, such as in yachts, cars, and airplanes, due to their high elastic properties and damping [89,125–129]. Pure luffa fibers have a significant sound absorption coefficient, which indicates that thick, pure luffa specimens can result in better acoustic properties [87]. The addition of luffa fibers into a polymeric matrix caused it to exhibit profusely high acoustic properties in the end NFC at all frequencies, since it improved the soundproofing to four times higher at moderate frequency levels [45]. Saygili et al. [130] observed that the damping level of luffa/epoxy was higher than that of jute/epoxy natural fiber composite. Furthermore, Genc et al. [89] concluded that the acoustic properties of LNFCs with an epoxy matrix were significantly promising, which can allow them to be involved in practical usages, while Shen et al. [77] recommended the utilization of LNFCs for sound absorption and vibration applications.

### *3.3. Mechanical Properties*

Mechanical characteristics of an NFC are specified through multiple tests, like hardness, flexural, tensile, compression, impact, etc. [131,132]. The mechanical properties of LNFCs are summarized and discussed in this section. Mechanical testing results obtained in recent LNFC studies are compiled and classified in Table 5, all the listed tests were conducted following the standard of American society for testing materials (ASTM).

**Table 5.** Mechanical properties of luffa natural fiber composites.

| Resin/Fiber Treatment | Fiber Size, Shape, and Composition Vf/wt.% | Hybrid/Filler | Tensile (MPa) | Flexural (MPa) | Compression (MPa) | Impact | Hardness | Other | Ref. |
|---|---|---|---|---|---|---|---|---|---|
| Epoxy/NaOH | Particles, short fibers, and mat shaped fibers (0.3–0.5 Vf) | | 23 | 115 | 107 | 27 KJ/m$^2$ | | | [59] |
| Epoxy/NaOH | 10–20 mm (10%, 20%, 30%, 40% and 50% Vf) | Ground nut (1:1) | 20 | 72 | 52.22 | 1.3 J | | | [62] |
| Epoxy/NaOH | Mat (30% Vf) | Flax | 24 | 59 | | 1.9 J | | | [49] |
| Epoxy/NaOH | | Ceramic B$_4$C (10 wt.%) | 13.56 E: 73.29 | | 34.39 | 18000 J | 91 HRc | | [44] |
| Epoxy | Rectangular mat (8, 13 and 19 wt.%) | | 18 E: 699 | 28 | | 4.90 KJ/m$^2$ | 217.20 MPa | Interlaminar shear stress (ILSS):1.38 MPa Erosion: 45% to 60% IA | [90] |
| Epoxy | Rectangular mat (19.87–30.86% Vf) | Glass fiber | 35.34 | 108.36 | | | 53.825 Hv | ILSS: 5.628 MPa | [91] |
| Epoxy/NaOH, and furfuryl alcohol followed by oxidation (sodium chlorite + acetic acid) | 100 × 100 mat (30 wt.%) | | 226.40 E:5865.70 | | | 7 KJ/m$^2$ | | | [50] |
| Epoxy/NaOH, benzoyl chloride, and potassium permanganate KMnO4 | Rectangular mat (13 wt.%) | | 28 E: 910 | 54 Ef: 3800 | | 7.3 KJ/m$^2$ | | ILSS: 2 MPa | [51] |
| Epoxy/NaOH | Chopped randomly (30%, 40% and 50% Vf) | | 18 | 55 | 105 | 0.68 J | | | [63] |

**Table 5.** *Cont.*

| Resin/Fiber Treatment | Fiber Size, Shape, and Composition Vf/wt.% | Hybrid/Filler | Tensile (MPa) | Flexural (MPa) | Compression (MPa) | Impact | Hardness | Other | Ref. |
|---|---|---|---|---|---|---|---|---|---|
| Epoxy/NaOH | 6 cm (30% Vf) | Silica nanoparticles | 13 E: 3284 | 28.9 | 81 | 0.9 J | | | [61] |
| Epoxy/NaOH, acetic acid | 2 mm (8, 9, 9.5 and 10 wt.%) | Lignite Fly Ash filler | 17.28 | 46.87 | 48.13 | | 98 RHN | | [54] |
| Epoxy | Rectangular mat (8, 13, and 19 wt.%) | | 16.76 | 24.825 | | | | ILSS: 1.38 MPa | [88] |
| Epoxy/NaOH | 2–5 mm (3.2–9.6 wt.%) | | E: 5560 | | | | | Vibration: 325.108 Hz | [25] |
| Epoxy | | Ceramic fibers | 140.68 | | 8.22 KN | 2 J | 77.3 RHN | | [84] |
| Epoxy/NaOH | 2 mm | Carbon fibers | 60.48 | 98.71 | 78.46 | | 92 RHN | | [133] |
| Polyester | 30 wt.% | Natural fillers (ground nut shell, rice husk, and wood powder) (3, 7 and11 WT.%) | 31.5 | | | 9 J | 13.3 Hv | Wear: 0.0284150 mm$^3$/Nm | [24] |
| Polyester/NaOH, hydrogen peroxide, acetic anhydride, and acetone | | | | 52.3 Ef: 3290 | | | | | [65] |
| Polyester/NaOH, meth-acrylamide | Short fibers and mat (24.5–42.6% Vf) | | 22 E: 5200 | | | 3.71 KJ/m$^2$ | | | [60] |
| Polyester | 20 mm (15–30% Vf) | E-glass | | 30.1 Ef: 1710 | | | | | [94] |
| Polyester | Mat (10, 20, 30 and 40 wt.%) | | | | | 1.12 J | | | [96] |
| Polyester/NaOH | Strips 120 × 25 mm (30% Vf) | | | 46.4 Ef: 3220 | | | | | [70] |

**Table 5.** *Cont.*

| Resin/Fiber Treatment | Fiber Size, Shape, and Composition Vf/wt.% | Hybrid/Filler | Tensile (MPa) | Flexural (MPa) | Compression (MPa) | Impact | Hardness | Other | Ref. |
|---|---|---|---|---|---|---|---|---|---|
| Polyester/methyl ethyl ketone peroxide | Mat 250 × 100 mm | | 23.893 | | | | | | [56] |
| Polyester/NaOH, hydrogen peroxide acetic anhydride, and acetone | Mat (5–15 wt.%) | | | 41.96 Ef: 2690 | | | | | [66] |
| RF/Calcium phosphate and calcium carbonate | 50 wt.% | | 14.88 E: 680 | 80.67 Ef: 3338 | 70.28 | | | | [78] |
| RF/NaOH, CaCl2, H2SO4, and Na2HPO4 | 2 cm (10–50 wt.%) | | 29.438 E: 1662 | | 81.00 | | | | [57] |
| Formaldehyde | Mat in medium density fireboard | Pine, beech and oak | | 50.91 | | | | | [97] |
| Polyurethane | Mat | Glass fiber | 12.7 | | | | | | [99] |
| PP/NaOH and silane coupling agent | 2–15 wt.% | | 35 | | | | | | [68] |
| Bio-based polyethylene (HDPE) | (10, 20, 30 and 40 wt.%) | | 21.2 E: 2082 | 37.7 | | 33.3 J/m | | | [98] |
| Vinyl ester/ethanol, NaOH, BTDA dianhydrides | Mat (15 wt.%) | | 21.2 | | | | | DMA: | [64] |
| Polylactic acid/NaClO | 2 cm (2, 5 and 10 wt.%) | | 36.44 E: 2997.45 | 48.64 Ef: 3939 | | 28.19 J/m | | | [58] |
| Geopolymer | Fiber's D: 200 μm. L: 160 mm. (10 vol.%) | | | 14.2 | 31 | | | | [100] |

**Table 5.** *Cont.*

| Resin/Fiber Treatment | Fiber Size, Shape, and Composition Vf/wt.% | Hybrid/Filler | Tensile (MPa) | Flexural (MPa) | Compression (MPa) | Impact | Hardness | Other | Ref. |
|---|---|---|---|---|---|---|---|---|---|
| Concrete | Mat | | | 10.2 | 25.8 | | | | [103] |
| Pre-gelatinized cassava starch | 3–5 mm (5, 10, 15, and 20 wt.%) | | 1.24 | | | | | | [101] |
| Pure luffa/NaOH, H2O2, CH3COOH, | 30–40 mm/Bundle | | 74.23 E: 820 | | | | | | [55] |
| Pure luffa/glacial acetic acid and hydrogen peroxide | 2 20 and 30 mm | | 91.63 E: 1897 | | | | | | [27] |
| Pure luffa/thermo-hydromechanical treatment | | | 177.93 | | | | | | |
| Pure Luffa | | | | | 0.74 | | | | [77] |
| Pure luffa | | | 120 E: 2300 | | 0.37 | | | | [47] |
| Pure luffa | Specimen's D: 42–81 mm | | | | 0.40 | | | | [46] |

### 3.3.1. Tensile Properties

Luffa natural fiber composite studies have mostly considered the tensile behavior compared to other mechanical properties, aiming to define the elastic behavior of this NFC, i.e., ultimate tensile strength as well as Young's modulus. Tensile strength (TS) of LNFCs increased by increasing fiber content up to 40 wt.%, but it started to decrease at 50 wt.% [59,62]. Thus, Mani et al. [134] reported an increment in the tensile strength of luffa mat/epoxy NFC followed by a decrement when adding fibers beyond 40 wt.%. However, by considering chopped luffa fibers, 50 wt.% revealed the highest strength. In a sandwich structure, the tensile strength of LNFC increased by increasing fiber content up to two luffa layers then decreased by adding a third layer [90]. Generally, LNFC made with NaOH-treated fibers exhibited higher tensile strength compared to that of untreated fibers [49,51,60]. Moreover, benzoyl treatment showed better property improvement than alkali and $KMnO_4$ treatments [51]; in addition, BTDA dianhydrides (tetracarboxylic benzophenone dianhydrides) revealed the best TS increment compared to ethanol and NaOH treatments [64]. However, Furfuryl Alcohol-grafted LNFC revealed a notable improvement in tensile strength (100%) and tensile modulus (123%), which were higher than the ones obtained by NaOH-treated LNFC [50]. Superior TS can be obtained by hybridizing LNFC with natural or synthetic fibers such as ceramic B4C, which improved TS around 8.6% [44], glass fiber (up to 100.4%) [88,91], and ground nut shell. In contrast, hybridization of LNFC with wood powder or rice husk reduced its tensile strength [24]. Adding silane coupling agent into luffa/polypropylene NFC can deteriorate its TS, whereas increasing the luffa fiber volume fraction reduced the tensile strength [68]. In single luffa fiber testing, 10% NaOH treatment for 30 min at 40 °C worked best in improving the fiber's tensile strength [55].

As clearly shown in Table 5 and Figure 6, peak tensile strength of LNFC was observed in an epoxy matrix, followed by polylactic acid and polypropylene. However, the highest LNFC tensile strength reported was 226 MPa in an epoxy matrix, followed by 140 MPa in a ceramic/luffa/epoxy hybrid NFC. Hence, LNFC with polylactic acid matrix reached a tensile strength of 36.44 MPa and a tensile modulus of around 3 GPa; thus, luffa in a polypropylene matrix exhibited 35 MPa TS. Luffa NFC revealed a tensile strength of 31.5 MPa with a polyester matrix, 29 MPa in resorcinol-formaldehyde, 21 MPa in bio-based high-density polyethylene as well as in vinyl ester, and lower tensile strength was observed in a polyurethane matrix (12.7 MPa). In terms of pure luffa, the peak tensile strength observed was 177.93 MPa after treating fibers with a thermo-hydro-mechanical technique. Paglicawan et al. [135] mentioned that the stress and strain curve of luffa/polyester NFC was similar to the curve of brittle materials, hence it behaves elastically as stress and stress increases in linear trend.

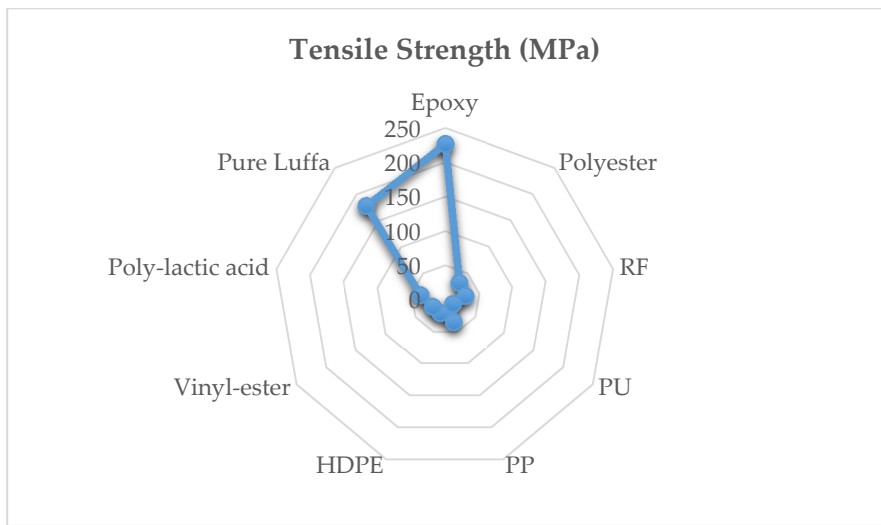

**Figure 6.** Tensile strength of luffa natural fiber composites.

### 3.3.2. Flexural Properties

Increasing fiber volume fraction enhanced the flexural strength of LNFC, as these fibers helped enduring bending loads throughout the brittle resin [59], yet adding fibers beyond 50 wt.% decreased the flexural strength of the end LNFC [62]. Chemically treating luffa fibers with NaOH, acetic anhydride, or acetone can result in higher flexural strength than those of untreated fibers [65,66]. Benzoyl treatment exhibited better bending properties enhancement, compared to KMnO4 and alkali treatments [51]. Hybridizing LNFC with flax fibers [49] or glass fibers [91] can reveal greater flexural strengths, especially when luffa is considered as a core layer in laminate structure [88]. However, the flexural strength of LNFCs increased by increasing fiber content up to two luffa layers, then showed a decrease by adding a third layer [90]. As we can see in Table 5, maximum flexural strength observed was 115 MPa with NaOH-treated luffa fibers and epoxy, followed by 108 MPa in glass/luffa/epoxy hybrid natural fiber composite. Moreover, selecting Resorcinol-Formaldehyde as a matrix produced a flexural strength of around 80 MPa, while luffa/polyester NFC reached 52.3 MPa flexural strength. Lower values were observed in LNFC with a geopolymer matrix (14.2 MPa) and concrete (10.2 MPa).

### 3.3.3. Compression Properties

In contrast with the aforementioned mechanical properties, compressive strength of luffa natural fiber composites decreased by increasing fiber composition (wt.%) [59,78]. Optimum compressive strength can be obtained in an NFC with a fiber volume fraction around 30% [62]. Chemically treated luffa fibers contributed to around 6% higher compressive strength [57], and adding nanofillers like ceramic B4C [44] or $SiO_2$ [61] can reveal superior compression strengths. Table 5 shows that the highest compressive strength reported was 107 MPa and 105 MPa in NaOH-treated luffa fibers and epoxy matrix, respectively. Furthermore, reinforcing resorcinol-formaldehyde with luffa chopped fibers reached a compressive strength of 81 MPa, while lower compressive characteristics were exhibited in an LNFC with a geopolymer matrix (31 MPa) and concrete matrix (around 29 MPa). Hence, highest compressive strength reported for pure luffa was 0.74 MPa.

### 3.3.4. Impact Properties

The toughness of LNFCs can be raised through increasing fiber volume fraction [24,50]. However, NagarajaGanesh et al. [96] indicated that 30 wt.% revealed the highest impact strength in luffa/polyester NFC, whereas Anbukarasi et al. [59] observed maximum toughness in samples with 40 wt.% treated mat/epoxy, as it showed 163.6% impact strength improvement. Moreover, chemical treatments enhanced the impact strength of luffa fibers [49,60]. Sodium hydroxide treatment had higher fiber toughness improvement compared with sodium chlorite and acetic acid [50]. Furthermore, fibers treated with $KMnO_4$, benzoyl chloride, and NaOH exhibited 36%, 49%, and 32% higher toughness, respectively, than untreated luffa fibers [51]. Some natural fibers and nanofillers showed their compatibility with LNFC, such as ground nut shells [62,96], and $B_4C$ that improved the toughness by ~54% [44]. It is worthy to mention that the impact strength can be highly affected by the fiber's shape and volume [90]. As Table 5 shows, the most significant impact strengths were observed in LNFC with an epoxy matrix.

### 3.3.5. Hardness

Few studies have investigated the hardness of LNFCs, aiming to determine the effect of layer number in a sandwich structure, or the difference after adding nanofillers or other natural fibers. Reinforcing epoxy matrix with two layers of luffa mat exhibited peak hardness, while adding a third layer led to hardness deterioration [90]. Addition of ceramic $B_4C$ nanoparticles contributed to increasing the hardness of LNFC around 8 HRC [44]. Similarly, hybridizing LNFC with ground nut shell improved its hardness in contrast with wood flour and rice husk, which reduced this property [24]. As illustrated in Table 5, peak hardness value reported was 98 RHN in treated luffa/epoxy NFC,

followed by 92 RHN in hybrid NaOH-treated luffa/carbon/epoxy hybrid NFC. Moreover, LNFC with a polyester matrix revealed a hardness of 13.3 Hv.

### 3.3.6. Wear, Shear, Vibration, and Dynamic Mechanical Analysis

A material is brittle or ductile due to its erosion rate versus angle of impingement. A material is classified brittle if the highest erosion happens at a 90° impact angle, while a material is considered ductile if the peak erosion rate occurs on a low angle of impingement (below 30°) [90]. In LNFCs, a ductile to brittle transition was observed by increasing the fiber content (from two layers to three layers); however, the impingement angle switched from 45° to 60° [90]. Regarding the abrasive wear of LNFC, increasing the fiber volume fraction increased the wear rate because of the degraded matrix/fiber adhesion [24]. Interlaminar shear stress (ILSS) describes the stresses affecting the interface of two plies of a composite material in a sandwich structure. Moreover, these stresses can deform the connection between these two plies and lead the composite to split if the stresses are adequate [91]. ILSS of NFCs exhibited a continuous increase simultaneous to fiber weight increment [88,90]. ILSS was enhanced around 95% by chemically treating luffa fibers with sodium hydroxide [51]. Regarding the vibration behavior of LNFCs, increasing the thickness ratio as well as the aspect ratio reduced the frequency response [25]. In terms of the dynamic mechanical behavior of LNFCs, exposing this composite material to a second heating cycle showed an increase in the storage modulus as well as the loss modulus. However, the improved surface of the chemically treated fibers contributed to enhancing the matrix/fiber bonding and thereby increased the loss and storage modulus [64]. However, Kalusuraman et al. [71] reported highest damping peak in samples with 50 wt.% luffa fibers treated with $CaOH_2$, and highest loss modulus in specimens with 50 wt.% silane-treated luffa fibers.

### 3.4. Chemical, Thermal, and Electrical Properties

Thermal properties of LNFC specify its behavior when exposed to high temperatures, which involves several core aspects like melting point, thermal degradation, and crystallinity degree. Thus, additional properties such as electrical and chemical are imperative in proving the reliability of LNFC. Lately, different tests have been conducted in the area of LNFCs, e.g., Fourier-transform infrared spectroscopy (FTIR), thermogravimetric analysis (TGA), chemical, differential scanning calorimetry, etc.

### 3.4.1. Fourier-Transform Infrared Spectroscopy

The general infrared spectrum properties of luffa fibers are primarily due to lignin, hemicelluloses, and $\alpha$-cellulose [136]. FTIR was carried out for LNFCs with an epoxy matrix [50,51,61], polyester [65,96], vinyl ester [64], resorcinol-formaldehyde [78], pre-gelatinized cassava starch [101], as well as pure luffa fiber [55,69]. FTIR results of the aforementioned LNFCs showed dissimilar spectra trends, as well as different peak numbers and positions, which was due to the distinct fiber treatment types, percentage, and methods

Fourier-transform infrared spectra researches confirmed the hydrophilic nature of luffa fibers [96]. Luffa fibers treated with NaOH, $KMnO_4$, and benzoyl chloride showed lower absorption spectra than untreated luffa fibers, due to the elimination of non-polar covalent components such as fat, wax, and so on. Moreover, benzoyl chloride chemical treatment exhibited the greatest spectra reduction compared to other chemical treatments, although after chemical modifications the main functional components of luffa fibers remained the same [51]. After treatment with sodium hydroxide, the band of pure luffa fibers at 1245 $cm^{-1}$ completely disappeared, and the band at 1375 $cm^{-1}$ significantly decreased, which showed that the partial acetyl elements in the hemicellulose and lignin dissolved in the alkaline [55]. However, addition of luffa fibers into a carbohydrate matrix like pre-gelatinized cassava starch (TPS) displayed negligible changes on the FTIR spectra due to the identical chemical structure of the matrix and the fibers. Hence, addition of luffa fibers slightly reduced the OH stretching peak of gelatinized cassava starch (from 3404 $cm^{-1}$ to 3395 $cm^{-1}$), which indicated the creation of hydrogen bonds between the gelatinized cassava starch and luffa fibers [101].

### 3.4.2. Chemical

Akgül et al. [97] tested the chemical behavior of luffa fibers, and the results showed that its chemical components consisted of 0.37% ash, 14.04% lignin, 62.34% α-cellulose, and 84.84% holocellulose. Moreover, luffa had the following solubility: 16.38% in 1% NaOH, 4.5% in cold water, 3.30% in hot water, and 0.25% in alcohol-benzene. However, luffa exhibited a solubility comparable to hardwood solubility when compared with other annual plants like hazelnut husk, cotton stalks, corn stalks, cotton carpel, cereal straw, softwood, and hardwood. Luffa revealed higher content of α-cellulose and holocellulose, whereas ash and lignin contents were lower than almost all compared plants [97].

### 3.4.3. Thermogravimetric

Thermogravimetric analysis (TGA) is a testing technique that indicates the mass loss of a material upon temperature exposure for a time period [137]. TGA was conducted for LNFCs with the following matrices: epoxy [50,51,59], polyester [94,96], TPS [101], vinyl ester [64], resorcinol-formaldehyde resin [78], and pure luffa [27,69]. Figure 7 shows a TGA of treated and untreated luffa fibers.

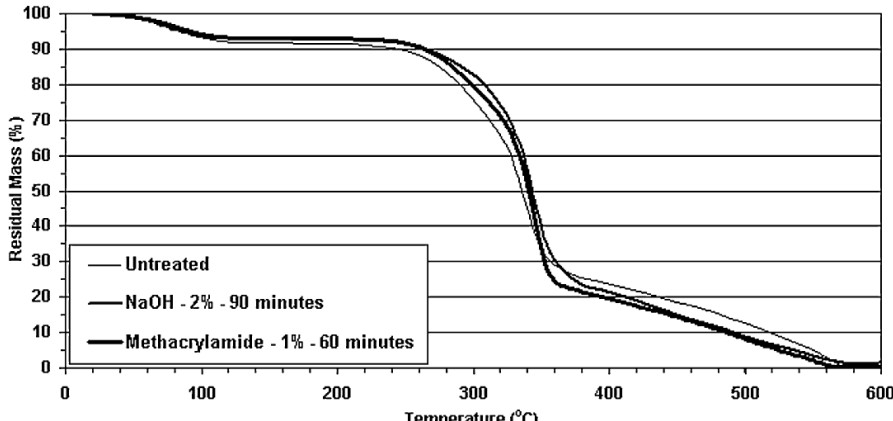

**Figure 7.** Thermogravimetric analyses of treated as well as untreated luffa fibers. Adapted with permission from ref. [69]. Copyright 2020, Elsevier.

Luffa fibers lost 7.25% of their weight in the temperature range between 49 °C and 118 °C due to the evaporation of water, then lost 46.5% between 118 °C and 322 °C due to the thermal oxidative degradation of lignin, cellulose, and hemicelluloses; hence, they lost 35.69% between 322 °C and 455 °C due to char creation. It is worth mentioning that hemicellulose decomposition occurred at a temperature ranging from 160 °C to 250 °C [96].

Thermal resistance of luffa fibers slightly increased when the fibers were treated. Temperatures beyond 250 °C caused the highest mass changes as these fibers reached a drastic level of thermal degradation [69]. However, the thermal stability of untreated luffa fibers was lower than that of chemically treated luffa fiber. Compared with benzoyl chloride and NaOH, KMnO$_4$ chemical treatment showed the best improvement in the thermal stability of luffa fibers [51]. Furthermore, adding luffa fibers to a matrix like TPS can enhance its thermal stability [101]. Thus, LNFC can endure temperatures up to 800 °C before its decomposition [78].

### 3.4.4. Differential Scanning Calorimetric

Differential scanning calorimetric (DSC) analysis was carried out by Tanobe et al. [60] on the polyester matrix to specify its curing temperature and curing time at a heating rate of 10 °C/min from 20 °C to 180 °C for each specimen. DSC results of pure polyester exposed to 70 °C showed that the curing began straight after combining the resin with hardener, and took around 10 min to complete

at 120 °C, whereas DSC results of untreated LNFC indicated that the resin was partially cured at 80 °C [60].

All luffa/epoxy NFCs revealed two transition peaks, an exothermic peak above 200 °C, and endothermic peak at ~100 °C. The endothermic peak of untreated LNFCs was observed at ~100 °C, caused by moisture vaporization from the fibers. However, in treated LNFCs, the aforementioned peak was observed at higher temperatures (FA-grafted NFC at 110 °C and NaOH-treated NFC at 106 °C). Furthermore, the curing reaction enthalpy (ΔHcure) of neat epoxy was slightly lower than that of LNFCs [50].

### 3.4.5. Dielectric

The dielectric behavior of LNFC revealed greater constants upon increasing the fiber volume fraction; for instance, the addition of heat-treated luffa fibers into a Polylactic acid matrix exhibited higher dielectric constants compared to pure PLA at all applied frequencies, yet increasing the frequency decreased the dielectric constants [80]. The interfacial polarization produced between the dipoles of the matrix and fibers was the main reason for the notable dielectric constant revealed in LNFC [138]. However, the hybrid luffa/coir NFC exhibited a dielectric constant similar to that of composite materials made with synthetic fibers [74].

## 4. Hybrid

The characteristics of hybrid LNFC depend on the failure strain of each fiber type considered, fiber orientation, fiber volume fraction, fiber length, fiber/matrix adhesion, fiber layup [139]. The highest properties can be reached once the strains of both hybridized fibers are notably compatible, property results [140]. Table 6 lists the major selected fibers utilized for hybridization of LNFC.

**Table 6.** Hybrid LNFCs.

| Resin/Coupling Agent/Hardener | Hybrid/Filler | Reference |
|---|---|---|
| Epoxy | Ground nut (1:1) | [62] |
| Epoxy | Flax | [49] |
| Epoxy | Jute | [93] |
| Epoxy | Ceramic $B_4C$ (10 wt.%) | [44] |
| Epoxy | Glass fiber | [91] |
| Epoxy | Glass fiber | [141] |
| Epoxy | Carbon fiber | [133] |
| Epoxy | Silica nanoparticles | [61] |
| Epoxy | Lignite fly ash filler | [54] |
| Epoxy | Lead oxide nanofiller | [73] |
| Epoxy | Coir | [74] |
| Polyester | Natural fillers (ground nut shell, rice husk, and wood powder) (3, 7, and 11 wt.%) | [24] |
| Polyester | E-glass | [94] |
| Polyester | $TiO_2$, $Al_2O_3$, and $CaCO_3$ | [142] |
| PU foams | Tea leaf (main in the study) | [45] |
| PU | Glass fiber | [99] |
| formaldehyde | Pine, beech, and oak | [97] |
| Poly (Butylene Succinate-Co-Lactate)/starch blends | Kenaf | [143] |

As shown in Table 6, mostly hybrid LNFCs were considered in an epoxy matrix, yet luffa/epoxy NFCs were hybridized with several natural fibers like ground nut, flax, jute, fly ash, coir, etc., and with different synthetic fibers, viz., glass and carbon fibers. Hence, other LNFCs with a polyester matrix were hybridized with wood powder, rice husk, ground nut shell, E-glass, and so on. Generally, hybridizing LNFC with other fibers reveals greater mechanical characteristics [49]. The addition of luffa and tea leaf fibers to polyurethane foam exhibited significant acoustic properties [45]. In terms

of the luffa/glass hybrid NFC, optimal flexural and tensile strengths were obtained by considering two glass fiber plies as outer layers and two luffa plies as a core [91]. Adding B4C into the luffa/epoxy NFC enhanced its tensile properties by 8.55% and compressive properties by 67%, yet reached a hardness of 91 HRC [44]. $SiO_2$ nanoparticles contributed to increasing the mechanical properties around 2% [61]. Similarly, fly ash filler was able to enhance the tensile strength of LNFC by improving the matrix-to-fiber adhesion [54]. However, in the luffa/polyester NFC, adding wood powder, rice husk, and ground nut shell improved the physical and mechanical properties [24].

## 5. Conclusions

The aim of this review paper is to compile, categorize, and compare the research in the area of LNFCs, taking into consideration their physical, morphological, mechanical, thermal, electrical, and acoustic properties, shedding light on the matrix types, fiber sizes, fiber volume fractions, and fiber treatments. Diverse chemical treatments were utilized to modify the characteristics of luffa fibers, e.g., sodium hydroxide, hydrogen peroxide, acetic acid, benzoyl chloride, acetone, HCl, etc. In addition, thermo-mechanical, thermo-hydro-thermal, and heat treatments were implemented. Luffa fibers were considered as a reinforcement for various types of matrixes, such as epoxy, polyester, formaldehyde, polylactic acid, high-density polyethylene, vinyl ester, polyurethane, and so on. Scanning electron microscopy showed that luffa has a flake-like, fatty, and waxy structure. Moreover, increasing the fiber-loading in LNFC reduced its crystallinity and increased the crystallite size. Regardless of the chosen matrix, LNFC densities ranged between 0.717 and 1.38 g/cm$^3$. However, addition of luffa fibers into polymeric matrices was able to increase the overall water absorption. LNFCs exhibited significant sound absorption coefficients throughout all fiber volume fractions. In terms of the mechanical behavior of LNFCs, the highest mechanical properties were observed through selecting epoxy resin as a matrix, for example, the highest tensile strength of 226 MPa, flexural strength of 115 MPa, compressive strength of 107 MPa, and hardness of 98 RHN. Furthermore, chemically treating these fibers showed a notable effect on the mechanical characteristics of the end LNFC. Hence, the chemical composition of luffa fibers exhibited higher holocellulose and $\alpha$-cellulose content, compared to other natural fibers, yet it had lower ash and lignin contents. Some hybrid LNFCs had dielectric constants similar to those of syntactic composite materials. Different natural and synthetic fibers were utilized to hybridize LNFC, such as flax, jute, coir, tea leaf, ground nut, glass, carbon, etc., which commonly contributed to exhibiting greater properties compared to pure LNFCs. Several researches proposed the involvement of LNFC in building, printed circuit boards (PCBs), and further applications. Acoustic, thermal, and chemical properties as well as the dimensional stability of LNFC still need more research. In addition, the utilization of LNFC in household goods, sports, automotive, aerospace, and other industrial areas needs further investigation. Furthermore, not many researchers have utilized design of experiments (DOE) to conduct experimentation. However, DOE is a useful scientific method that helps plan and organize experiments in order to reach valid conclusions through analyzing the obtained results. In other words, it reduces the time and resources to make experiments, eliminates redundant observations, and provides points that can be used to create a meta-model by involving a smart exploration of the design space. Moreover, machine learning techniques like Artificial Neural Networks (ANNs) are widespread approximators that are usually involved in regression tasks and classification; thus, these techniques are highly capable of approximating the properties of LNFC.

Future research in the LNFC area needs to focus on the consideration of polymeric recycled matrices, the modeling and simulation of LNFCs, the analytical analysis of LNFC, such as using Halpin–Tsai or Cox models to predict the tensile strength, Hirsch's model for Young's modulus, Hashin's approach for fatigue criteria, the Johnson–Champoux–Allard technique to analyze the sound absorption, and Fourier's heat conduction equation to predict the thermal characteristics, in addition to the implementation of optimization algorithms in order to get optimal parameters, and the design of experiment approaches for the variation analysis.

**Author Contributions:** All authors have read and agreed to the published version of the manuscript. M.A. (Mohamad Alhijazi) was responsible for Data curation, Resources, Visualization. B.S. was responsible for Investigation. Q.Z. was responsible for Supervision, Methodology. B.S., Q.Z., M.A. (Mohammed Asmael), M.A. (Mohamad Alhijazi), A.E. and Z.Q. were responsible for Writing—original draft and Writing—review & editing. B.S. and Q.Z. were responsible for Project administration. All authors have read and agreed to the published version of the manuscript.

**Funding:** This research received no external funding.

**Conflicts of Interest:** The authors declare no conflict of interest.

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
