# Peer review of "Recent Developments in Luffa Natural Fiber Composites: Review"

_sustainability, doi:10.3390/su12187683_

Round 1

Reviewer 1 Report

This is a good review paper and the presented results might broaden our understanding of the natural fiber-reinforced composites. This work could offer valuable references for composite design and property optimization. The novelty of this work, as well as its interesting results, are clearly described in the article. Therefore, the referee would like to recommend this work to be published in the journal. However, there are a few issues to be addressed before its publication.
1. The authors should discuss the results more.
2. The abstract should be improved.
3. I would recommend them to summarize which analytical method is more suitable for a specific property of Luffa Natural Fibers Composites, such as tensile, flexural, thermal, etc…, in the appropriate place of the manuscript to improve their literature.
4. The conclusion part can be enhanced and improved.

Author Response

Response To Reviewer #1

Manuscript ID: Sustainability-913258

Recent Developments in Luffa Natural Fibers Composites: Review

Mohamad Alhijazi, Babak Safaei, Qasim Zeeshan, Mohammed Asmael, Arameh Eyvazian and Zhaoye Qin

Dear Editor:

We wish to thank you for the timely processing of our article in “Sustainability” and the reviewers for their constructive comments, which have helped us to clarify a number of issues in our manuscript. We have attended to all the raised points/concerns in this revised manuscript. Given below is our response and the suggested revisions to the article, as highlighted in the manuscript.

Sincerely yours,

Dr. Babak Safaei

PhD (Tsinghua University)

Assistant Professor of Mechanical Engineering

Eastern Mediterranean University

Dated: 04.09.2020

Reviewer’s comments:

Reviewer #1

This is a good review paper and the presented results might broaden our understanding of the natural fiber-reinforced composites. This work could offer valuable references for composite design and property optimization. The novelty of this work, as well as its interesting results, are clearly described in the article. Therefore, the referee would like to recommend this work to be published in the journal. However, there are a few issues to be addressed before its publication.

  • The authors should discuss the results more.

Response:

The following is modified/ included at Page No. _12_ Line no. _298-302_ Page No. _13_ Line no. __320-323 &331-334_and Page No. _14_ Line no. __ 357_ (Highlighted in Green)

Peak tensile strength of LNFC was observed in Epoxy matrix, followed by Polylactic acid and Polypropylene. However, highest Luffa NFC tensile strength reported was 226 MPa in epoxy matrix, followed by 140 MPa in ceramic/Luffa/epoxy hybrid NFC. Hence LNFC with Polylactic acid matrix reached a tensile strength of 36.44 MPa and a tensile modulus of around 3 GPa, thus Luffa in polypropylene matrix exhibited 35 MPa TS. Whereas, Luffa NFC revealed a tensile strength of 31.5 MPa with polyester matrix, 29 MPa in resorcinol formaldehyde, 21 MPa in bio-based high-density polyethylene as well as in vinyl ester, and lower tensile strength was observed in polyurethane matrix (12.7 MPa).

Moreover, selecting RF as a matrix performed a flexural strength of around 80 MPa, whilst Luffa/polyester NFC reached 52.3 MPa flexural strength. Lower values were observed in LNFC with geopolymer matrix (14.2 MPa) and Concrete (10.2 MPa).

Furthermore, reinforcing resorcinol formaldehyde with Luffa chopped fibers reached a compressive strength of 81 MPa, while lower compressive characteristics were exhibited in LNFC with geopolymer matrix (31 MPa), and concrete matrix (around 29 MPa). Hence, highest compressive strength reported for pure Luffa was 0.74 MPa.   

Moreover, Luffa natural fiber composite with polyester matrix revealed a hardness of 13.3 Hv.

  • The abstract should be improved.

Response:

The following is modified/ included at Page No. _1_ Line no. _13-30_ (Highlighted in Green)

Natural fibers composites (NFC) is an evolving area in polymers sciences. Fibers extracted from natural sources hold a wide set of advantages such as negligible cost, significant mechanical characteristics, low density, high strength to weight ratio, environmental friendly, recyclable, etc. Luffa Cylindrica, also termed as Luffa Gourd or Luffa Sponge is a natural fiber that has a solid potential to replace synthetic fibers in composite materials for diverse utilizations, like vibration isolation, sound absorption, packaging and so on. Recently, many researches involved Luffa fibers as a reinforcement in NFC’s development, aiming to investigate their performance in the selected matrix as well as the behavior of the end NFC. This paper presents a review on the recent developments in Luffa natural fibers composites. Physical, morphological, mechanical, thermal, electrical, and acoustic properties of Luffa NFCs are compiled, categorized, and compared, taking into consideration the selected matrixes, fibers’ size, fibers’ volume fraction, and fibers’ treatments. Although Luffa natural fibers composites revealed promising properties, yet the addition of these natural fibers increases the water absorption. Moreover, chemical treatments like sodium hydroxide (NaOH) and Benzoyl can remarkably enhance Luffa fibers’ surface, remove undesirable impurities, and reduce the water uptake, thereby improve the overall characteristics. Hybridizing Luffa NFC with another natural fiber or synthetic fiber i.e. Glass, Carbon, Ceramic, Flax, Jute, etc. can contribute in enhancing the properties of the end composite material. However, Luffa fibers have exhibited a profuse compatibility with epoxy matrix.

  • I would recommend them to summarize which analytical method is more suitable for a specific property of Luffa Natural Fibers Composites, such as tensile, flexural, thermal, etc…, in the appropriate place of the manuscript to improve their literature.

Response:

The following is modified/ included at Page No. _18_ Line no. _508-512_ (Highlighted in Green)

The analytical analysis of Luffa NFC, such as using Halpin-Tsai or Cox models to predict the tensile strength, Hirsch’s model for young’s modulus, Hashin’s approach for fatigue criteria, Johnson-Champoux-Allard technique to analyze the sound absorption, and Fourier’s heat conduction equation to predict the thermal characteristics.

  • The conclusion part can be enhanced and improved.

Response:

The following is modified/ included at Page No. _18_ Line no. _483-514_ (Highlighted in Green)

The aim of this review paper is to compile, categorize and compare the research in Luffa natural fibers composites area, taking into consideration their physical, morphological, mechanical, thermal, electrical, and acoustic properties. Shedding the light on the matrix types, fibers’ size, fibers’ volume fraction, and fibers’ treatments. However, diverse chemical treatments were utilized to modify the characteristics of Luffa fibers i.e. Sodium Hydroxide, Hydrogen Peroxide, Acetic Acid, Benzoyl Chloride, Acetone, HCl, etc. Not only, thermo-mechanical, thermo-hydro-thermal and heat treatments were implemented as well. Luffa fibers were considered as a reinforcement for various types of matrixes, such as epoxy, polyester, formaldehyde, polylactic acid, high density polyethylene, vinyl ester, polyurethane and so on. Hence, scanning electron microscopy showed that Luffa has a flake-like, fatty and waxy structure. Moreover, increasing the fiber loading in Luffa NFC reduces its crystallinity and increases the crystallite size. Regardless of the chosen matrix, Luffa natural fibers composites densities ranged between 0.717 and 1.38 g/cm3. However, addition of Luffa fibers into polymeric matrixes was able to increase the overall water absorption. Luffa natural fibers composites exhibited significant sound absorption coefficients throughout all fibers volume fractions. While in terms of the mechanical behavior of Luffa NFC, highest mechanical properties were observed through selecting epoxy resin as a matrix, for example: highest tensile strength of 226 MPa, flexural strength of 115 MPa, compressive strength of 107 MPa, and hardness of 98 RHN. Furthermore, chemically treating these fibers showed a notable effect on the mechanical characteristics of the end LNFC. Hence, chemical composition of Luffa fibers exhibited higher holocellulose and α-cellulose content compared to other natural fibers, yet it has lower ash and lignin contents. Some hybrid Luffa NFCs are having dielectric constants similar to that of syntactic composite materials. Different natural and synthetic fibers were utilized to hybridize Luffa NFC, such as Flax, Jute, Coir, Tea Leaf Ground Nut, Glass, Carbon, etc. which commonly contributed in exhibiting greater properties compared to pure Luffa NFCs. Several researches proposed the involvement of Luffa NFC in building, electric board (PCB) and further applications. Future research in Luffa NFC area needs to focus on the consideration of polymeric recycled matrixes, the modeling and simulation of Luffa NFCs, the analytical analysis of Luffa NFC, such as using Halpin-Tsai or Cox models to predict the tensile strength, Hirsch’s model for young’s modulus, Hashin’s approach for fatigue criteria, Johnson-Champoux-Allard technique to analyze the sound absorption, and  Fourier’s heat conduction equation to predict the thermal characteristics. In addition to the implementation of optimization algorithms in order to get optimal parameters, and design of experiment approaches for the variation analysis.

Best Regards,

Dr. Babak Safaei

Reviewer 2 Report

The authors of this document did an extensive study and provided a valuable compilation and categorization of the research on Luffa based composites.

It would be valuable to spent more time on research gaps (for example swelling is hinted at, but its implication on results and need for further understanding can be stressed more) and critically discuss the found information to help readers with making the best choice for their experimentation/development.

The chosen form makes it difficult to compare results and needs improving (for example NaOH treatment comes back in several chapters, but the reader will struggle to have the complete overview of the impact on the complete material property profile).

Natural fibers like luffa struggle from non-uniform fiber orientation, which will impact mechanical properties in xyz direction. Add analysis on this.

Natural fibers in general have fluctuations in composition and property, luffa fibers are no exception. Add more background on the fluctuations in properties and their impact on final property (or selection criteria). 

small observations:

Figure 1 uses loofah instead luffa, to my knowledge they are the same, please replace for clarity. If not add luffa to the figure. 

More background can be given on the selection criteria for luffa, why would somebody work with this material vs. other available natural fibers, and for completeness also versus man-made fibers.

Table 1 has more precise numbers as quoted in the text, explain in the text.

Table 2 would benefit from listing the impact on water uptake, add a column with pro and/or cons.

It is mentioned that Methacrylamide harms the fibers, meaning?

After table 3 volume fraction is introduced, but from the text it is not clear whether volume fraction refers to matrix or filler.

Rectangular mat is mentioned as most used form (in addition to fibers, chopped materials), more words can be spent on differences in available raw material and its basic properties (table 1 hints at the ranges of physical properties, but does not detail to which shape they belong).

On several locations the authors refer to “…and other tests” “..and so on”..for completeness list all, or explain the field (....and other suitable tests for determining xyz)

3.2.3 can stress more the importance on thickness swelling and its impact on other characteristics.

Table 5 uses different units for impact, can it be standardized to a single unit for ease of comparison? also testing conditions are not included (dried sample, temperature of measurement, dimensions, etc) which creates the risk of comparing incomparable.

Would combination of table 4 and 5 be possible.

Figure 7; what is the meaning of the additional blue dots, does it refer to unused data or outliers?

The chemical composition in chapter 3.4.2 does not match table 1.

Take a critical look at self-citation.

Author Response

Response To Reviewer #2

Manuscript ID: Sustainability-913258

Recent Developments in Luffa Natural Fibers Composites: Review

Mohamad Alhijazi, Babak Safaei, Qasim Zeeshan, Mohammed Asmael, Arameh Eyvazian and Zhaoye Qin

Dear Editor:

We wish to thank you for the timely processing of our article in “Sustainability” and the reviewers for their constructive comments, which have helped us to clarify a number of issues in our manuscript. We have attended to all the raised points/concerns in this revised manuscript. Given below is our response and the suggested revisions to the article, as highlighted in the manuscript.

Sincerely yours,

Dr. Babak Safaei

PhD (Tsinghua University)

Assistant Professor of Mechanical Engineering

Eastern Mediterranean University

Dated: 04.09.2020

Reviewer’s comments:

Reviewer #2

The authors of this document did an extensive study and provided a valuable compilation and categorization of the research on Luffa based composites.

  • It would be valuable to spent more time on research gaps (for example swelling is hinted at, but its implication on results and need for further understanding can be stressed more) and critically discuss the found information to help readers with making the best choice for their experimentation/development.

Response:

The following is modified/ included at Page No. _12_ Line no. _298-302_ Page No. _13_ Line no. __320-323 &331-334_and Page No. _14_ Line no. __357_ (Highlighted in Green) and Page No. _9_ Line no. __246-249_ (Highlighted in Blue)

Hence LNFC with Polylactic acid matrix reached a tensile strength of 36.44 MPa and a tensile modulus of around 3 GPa, thus Luffa in polypropylene matrix exhibited 35 MPa TS. Whereas, Luffa NFC revealed a tensile strength of 31.5 MPa with polyester matrix, 29 MPa in resorcinol formaldehyde, 21 MPa in bio-based high-density polyethylene as well as in vinyl ester, and lower tensile strength was observed in polyurethane matrix (12.7 MPa).

Moreover, selecting RF as a matrix performed a flexural strength of around 80 MPa, whilst Luffa/polyester NFC reached 52.3 MPa flexural strength. Lower values were observed in LNFC with geopolymer matrix (14.2 MPa) and Concrete (10.2 MPa).

Furthermore, reinforcing resorcinol formaldehyde with Luffa chopped fibers reached a compressive strength of 81 MPa, while lower compressive characteristics were exhibited in LNFC with geopolymer matrix (31 MPa), and concrete matrix (around 29 MPa). Hence, highest compressive strength reported for pure Luffa was 0.74 MPa.   

Moreover, Luffa natural fiber composite with polyester matrix revealed a hardness of 13.3 Hv.

However, the swelling rate is low throughout the early moisture absorption stages of a natural fibers composite made with polymeric matrix, due to the viscoelastic behavior of this polymer. Hence, reaching a high swelling rate leads to the deterioration of the mechanical properties, such as tensile, compression, impact, bending, etc.  

  • The chosen form makes it difficult to compare results and needs improving (for example NaOH treatment comes back in several chapters, but the reader will struggle to have the complete overview of the impact on the complete material property profile).

Response:

Thank you for your valuable comment. As the paper is divided into sub-sections that discuss all properties of Luffa NFC, introducing a method that can improve the discussed property can be a valuable information for the scientists and the engineers, however sodium hydroxide (NaOH) proved its positive impact on a variety of natural fibers.

  • Natural fibers like luffa struggle from non-uniform fiber orientation, which will impact mechanical properties in xyz direction. Add analysis on this.

Response:

Thank you for your valuable comment. Since all natural fibers are Orthotropic, some researchers studied the effect of fibers’ orientation to determine their effect on the end NFC, yet in Luffa NFC field there is not such type of investigations, apparently due to the net-structure of Luffa plant, which makes it way too hard to extract long fibers and control their orientation in the developed NFC.

  • Natural fibers in general have fluctuations in composition and property, luffa fibers are no exception. Add more background on the fluctuations in properties and their impact on final property (or selection criteria).

Response:

Thank you for your valuable comment. In other NFC fields like Palm NFCs, very few researches studied the properties’ difference of palm fibers, after testing 4 samples obtained from diverse regions, trying to display the difference in the chemical composition as well as the characteristics. However, to date, there are no similar studies in Luffa NFC field, otherwise it would be of a great value to be added in our review paper.

  • Figure 1 uses loofah instead luffa, to my knowledge they are the same, please replace for clarity. If not add luffa to the figure.

Response:

The following is modified/ included at Page No. _2_ Figure no. _1_ (Highlighted in Blue)

  • More background can be given on the selection criteria for luffa, why would somebody work with this material vs. other available natural fibers, and for completeness also versus man-made fibers.

Response:

The following is modified/ included at Page No. _2_ Line no. _64-67_ (Highlighted in Blue)

Luffa fruit comprise light-weight natural fibers that have the potential to be utilized in reinforcing light-weight composites due to their polyporous structure, abundance, cheap price, as well as their surface morphology that can provide a good adhesion with the matrix.

  • Table 1 has more precise numbers as quoted in the text, explain in the text.

Response:

The following is modified/ included at Page No. _3_ Line no. _90-92_ (Highlighted in Blue)

As table 1 shows, Luffa density vary from 0.56 to 0.92 g/cm3, it has an average diameter of ~270 μm, and its microfibrillar angle is around 12 ⁰C. The chemical composition of Luffa consists of 63% cellulose, 20.88% hemi-cellulose, 11.69% Lignin, and 0.4% Ash.

  • Table 2 would benefit from listing the impact on water uptake, add a column with pro and/or cons.

Response:

Thank you for your valuable comment. Basically, Table 2 introduces the treatments that can be/were implemented to improve the properties of Luffa fibers, hence, all the listed treatments reduced the water absorption of Luffa fibers (more or less), whereas further explanations are following the table. Thank you once again.

  • It is mentioned that Methacrylamide harms the fibers, meaning?

Response:

The following is modified/ included at Page No. _4_ Line no. _109 &110_ (Highlighted in Blue)

Contrary to other chemical solutions, Methacrylamide treatment causes a serious deterioration in Luffa fibers’ integrity

  • After table 3 volume fraction is introduced, but from the text it is not clear whether volume fraction refers to matrix or filler.

Response:

The following is modified/ included at Page No. _5_ Line no. _132 & 133_ (Highlighted in Blue)

Luffa NFC studies involved diverse polymeric matrices with different weight ratios (Fibers’ Volume Fraction) which ranged from 2 to 50 wt%, however, most common weight composition was 30 wt%

  • Rectangular mat is mentioned as most used form (in addition to fibers, chopped materials), more words can be spent on differences in available raw material and its basic properties (table 1 hints at the ranges of physical properties, but does not detail to which shape they belong).

Response:

Thank you for your valuable comment. Naturally, Luffa fruit has a net-like structure, which therefore can be utilized as a mat for developing Luffa NFC, however some researchers preferred to crush/ chop these mats into smaller fibers in order to investigate the effect of fibers’ size on the overall properties of the final Luffa NFC.

In addition, Table 1 illustrates the physical and chemical properties of pure Luffa fibers, which generally belongs to all shapes of Luffa fibers.

  • On several locations the authors refer to “…and other tests” “..and so on”..for completeness list all, or explain the field (....and other suitable tests for determining xyz)

Response:

Thank you for your valuable comment. The aforementioned words “…and other tests” “..and so on” are considered similar to “etc”, which are utilized for exiting the future readers of this paper to discover more while reading the corresponding section/subsection. Furthermore, in the introduction of section 3.1, “Scanning Electron Microscope (SEM), transmission electron microscopy, polarized light microscopy, X-ray diffraction [113] and other tests i.e. were utilized”, the word “other tests” was utilized to keep a probability that there might be other Morphological test conducted in a paper that did not appear while we were searching for articles in this area, thereby was not included in our review paper.

  • 2.3 can stress more the importance on thickness swelling and its impact on other characteristics.

Response:

The following is modified/ included at Page No. _9_ Line no. _246-249_ (Highlighted in Blue)

However, the swelling rate is low throughout the early moisture absorption stages of a natural fibers composite made with polymeric matrix, due to the viscoelastic behavior of this polymer. Hence, reaching a high swelling rate leads to the deterioration of the mechanical properties, such as tensile, compression, impact, bending, etc.  

  • Table 5 uses different units for impact, can it be standardized to a single unit for ease of comparison? also testing conditions are not included (dried sample, temperature of measurement, dimensions, etc) which creates the risk of comparing incomparable.

Response:

Thank you for your valuable comment. In table 5, there are more than one unit for impact strengths due to the type of tests conducted (Charpy or Izod), hence converting these results to a single unit will be imprecise, which therefore will be unreliable for a comparison.

Moreover, in Natural fibers composite’s field, almost all researchers are conducting their tests following ASTM standard, for example ASTM D638/ D3039 for tensile testing, ASTM D256 for Impact, ASTM D790 for Flexural and so on.

The following is clarified at Page No. _9_ Line no. _269_ (Highlighted in Blue)

Mechanical testing results obtained in recent Luffa NFC studies are compiled and classified in Table 5, all the listed tests were conducted following the ASTM standard.

  • Would combination of table 4 and 5 be possible.

Response:

Thank you for your valuable comment, your suggestion is much appreciated. Table 4 shows the results of density tests of Luffa NFCs, which is considered as a physical property, yet Table 5 lists all the mechanical properties’ results. However, these two tables belong to two diverse sections. Thank you once again.

  • Figure 7; what is the meaning of the additional blue dots, does it refer to unused data or outliers?

Response:

Thank you for your valuable comment. The blue dots are the main values of this Radar chart, for example, the blue dot of Epoxy is beyond the range 200 MPa compared with that of Polypropylene (PP) which is below 50 MPa. Hence, this chart is providing a comparison of the tensile strengths of Luffa NFCs made with various matrixes’ types.

  • The chemical composition in chapter 3.4.2 does not match table 1.

Response:

Thank you for your valuable comment. Table 1 includes the general chemical composition of Luffa fibers, whereas in section 3.4.2. the values are the results of a chemical test conducted by Akgül et al.

The following is clarified at Page No. _15_ Line no. _410 & 411_ (Highlighted in Blue)

Akgül et al. [101] tested the chemical behavior of Luffa fibers, results showed that its chemical components consisted of: 0.37% ash, 14.04% lignin, 62.34% α-cellulose, and 84.84% holocellulose.

  • Take a critical look at self-citation.

Response:

Thank you for your valuable comment. We modified our manuscript.

Best Regards,

Dr. Babak Safaei

Round 2

Reviewer 2 Report

dear authors, thank you for the improved document.

There still is opportunity to improve the overall content of the document by putting more attention on knowledge gaps and identifying areas of research need based on the review you so carefully carried out. 

Author Response

Response To Reviewer #2

Manuscript ID: Sustainability-913258

Recent Developments in Luffa Natural Fibers Composites: Review

Mohamad Alhijazi, Babak Safaei, Qasim Zeeshan, Mohammed Asmael, Arameh Eyvazian and Zhaoye Qin

Dear Editor:

We wish to thank you for the timely processing of our article in “Sustainability” and the reviewers for their constructive comments, which have helped us to clarify a number of issues in our manuscript. We have attended to all the raised points/concerns in this revised manuscript. Given below is our response and the suggested revisions to the article, as highlighted in the manuscript.

Sincerely yours,

Dr. Babak Safaei

PhD (Tsinghua University)

Assistant Professor of Mechanical Engineering

Eastern Mediterranean University

Dated: 09.09.2020

Reviewer’s comments:

Reviewer #2

Dear authors, thank you for the improved document.

  1. There still is opportunity to improve the overall content of the document by putting more attention on knowledge gaps and identifying areas of research need based on the review you so carefully carried out.

Response:

Thank you for your valuable comment. We modified our manuscript.

Best Regards,

Dr. Babak Safaei
